# Inferring Drug-Protein–Side Effect Relationships from Biomedical Text

**DOI:** 10.3390/genes10020159

**Published:** 2019-02-19

**Authors:** Min Song, Seung Han Baek, Go Eun Heo, Jeong-Hoon Lee

**Affiliations:** 1Department of Library and Information Science, Yonsei University, Seoul 03722, Korea; goeun.heo@yonasei.ac.kr; 2Institute of Convergence, Yonsei University, Seoul 03722, Korea; seunghan.baek@yonsei.ac.kr; 3Department of Creative IT Engineering, POSTECH, Pohang 37673, Korea; jhlee@dblab.postech.ac.kr

**Keywords:** Biomedical Text Mining, semantic relatedness, Inference of Drug-Protein-Side Effect Relation

## Abstract

*Background*: Although there are many studies of drugs and their side effects, the underlying mechanisms of these side effects are not well understood. It is also difficult to understand the specific pathways between drugs and side effects. *Objective*: The present study seeks to construct putative paths between drugs and their side effects by applying text-mining techniques to free text of biomedical studies, and to develop ranking metrics that could identify the most-likely paths. *Materials and Methods*: We extracted three types of relationships—drug-protein, protein-protein, and protein–side effect—from biomedical texts by using text mining and predefined relation-extraction rules. Based on the extracted relationships, we constructed whole drug-protein–side effect paths. For each path, we calculated its ranking score by a new ranking function that combines corpus- and ontology-based semantic similarity as well as co-occurrence frequency. *Results*: We extracted 13 plausible biomedical paths connecting drugs and their side effects from cancer-related abstracts in the PubMed database. The top 20 paths were examined, and the proposed ranking function outperformed the other methods tested, including co-occurrence, COALS, and UMLS by P@5-P@20. In addition, we confirmed that the paths are novel hypotheses that are worth investigating further. *Discussion*: The risk of side effects has been an important issue for the US Food and Drug Administration (FDA). However, the causes and mechanisms of such side effects have not been fully elucidated. This study extends previous research on understanding drug side effects by using various techniques such as Named Entity Recognition (NER), Relation Extraction (RE), and semantic similarity. *Conclusion*: It is not easy to reveal the biomedical mechanisms of side effects due to a huge number of possible paths. However, we automatically generated predictable paths using the proposed approach, which could provide meaningful information to biomedical researchers to generate plausible hypotheses for the understanding of such mechanisms.

## 1. Introduction

Minimizing drug side effects is a key focus of medical treatment as well as future drug development. The side effects of drugs are well described in many clinical trials; however, these studies simply report on a given drug’s side effects and not attempt to explain the cause [1,2,3]. Because many factors are likely to connect a drug and a particular side effect, time and effort is required to discover the relationship between the two. Text mining approaches have been proposed as way to examine this relationship.

While many text mining studies have extracted overall relationships between drugs and side effects, they could not suggest the series of mechanistic steps from a drug to a particular side effect [4,5,6,7,8]. Therefore, we attempted to use text mining to identify concrete paths from a given drug and side effect. That is, if a drug and side effect are given as start and end points, respectively, the series of mediators (i.e., proteins) are extracted automatically from the text and connected to form a proposed drug-protein–side effect path.

Although there are public databases of pathways such as the KEGG (Kyoto Encyclopedia of Genes and Genomes) database, these databases have limited information and are not specialized for specific drug side effects. Using the Swanson ABC model as our foundation, we extracted the relation between entities and connected them by using the text mining method to study the extracted paths. There are several studies utilizing text mining to discover relationships between drugs and their side effects. Sohn et al. [9] proposed the decision tree approach, a machine learning algorithm, to extract sentences that contain drug and side effect pairs. To build feature sets, they employed side effect keyword features and pattern matching rules. Zhang et al. [10] developed a similarity driven matrix factorization method to predict the drug and side effect associations. The primary goal of their approach was to estimate the drug-side effect association relationship by identifying latent features for drugs and side effects to compute drug feature-based similarities and disease semantic similarity. Pauwels et al. [11] proposed a sparse canonical correlation analysis (SCCA) to predict potential side-effect profiles of candidate drugs based on their chemical structures. The primary goal of their approach was to extract correlated sets of chemical substructures and side-effects. Previous studies on the relationship between drugs and side effects primarily focused on the direct link between these two types of entities. Unlike these studies, the proposed approach is based on the indirect link between drugs and side effects via proteins.

In the present study, we used advanced text-mining techniques to extract relations between entities related to cancer drug research and to form a network with which to identify the chain reaction that occurs in the body when a drug is given [12,13,14,15,16,17,18]. We derive this network based from the interactions of entities suggested in scientific papers [19,20,21,22,23,24]. It is an accumulated and integrated network that connects fragmentary relations between drug-protein, protein-protein, and protein–side effect, thereby connecting drugs to their side effects [25,26,27,28].

Based on this composed knowledge network, we analyzed paths between drugs and side effects and ranked them using a new ranking function that combines semantic similarity scores and the frequency of entity pair co-occurrence to suggest meaningful paths. We compared our proposed ranking algorithm with other algorithms such as Correlated Occurrence Analogue to Lexical Semantics (COALS), and ontology-based algorithm (UMLS; Unified Medical Language System), and showed that the proposed algorithm outperforms the other three in terms of precision of K measurements.

Analyzing 20 identified paths, we were able to adjudge 65% of them as biologically plausible. Although our research does not show perfect performance, it can help to identify the link between drugs and side effects, and serve as a foundation for further research. Moreover, our method can be applied to other types of paths, thereby revealing previously unknown links among biological entities such as diseases and genes.

## 2. Materials and Methods

To investigate possible drug-protein–side effect paths, we propose a three-step framework: (1) pre-processing, (2) named entity recognition (NER) and relation extraction, and (3) path analysis (illustrated in Figure 1). Most cancer drugs target specific paths, which are composed of protein interactions. We therefore established proteins as our middle node. The first component is pre-processing, including parsing XML records that were downloaded manually from the PubMed search engine. During the parsing of records, we extracted important metadata such as PMID, title, and abstract. Once parsing was done, an abstract of the record was split into sentences by a sentence boundary detection algorithm; we used the algorithm provided in Stanford CoreNLP [29]. The second component has something to with NER and relation extraction. For NER, we employed a dictionary-based entity extraction. Since an entity is either a single word or a phrase, we needed to tokenize sentences by N-gram. N-gram denotes a contiguous sequence of n tokens. Once entity extraction is done, extracted entities and a sentence are fed into the relation extraction module. In relation extraction, Part-Of-Speech (POS) tagging was applied to the sentence to detect verbs. Since not every verb is meaningful in the bio-medical domain, we adopted the list of biomedical verbs provided in [10]. The third component was path analysis. With extracted entities and their relations, built a graph and generated k-depth paths. Once paths were generated, we computed semantic relatedness scores of any given pair of entities linked on the graph. To this end, we utilized two different sources: UMLS and the collected dataset a.k.a. corpus. UMLS was used to compute the similarity between two entities based on the concept hierarchical structure of UMLS. The corpus was used to build a corpus-based semantic relatedness model so that we could compute the similarity of two entities found in the model. To make it easy to examine the results reported by the proposed approach and as a utility function for the present paper, we developed a simple JSP (Java Server Page)-based web server that allowed us to navigate the top 350 paths. The web server is publicly accessible at http://informatics.yonsei.ac.kr:8080/drug_se/searchDrug.jsp. The web server provides a function of search by drug. The matched search result displays paths starting with drug, along with a PubMed ID that takes the reader to the PubMed record page. The results page also provides a link to the PubChem site for a particular drug. In addition, each path is broken down into a set of pairs such as drug and protein, protein and disease, etc. It is also displayed that a sentence of a PubMed record contains the particular pair in the path.

### 2.1. Data Source and Pre-Processing

We first select a specific domain related to cancer and retrieve relevant data from PubMed database (http://www.ncbi.nlm.nih.gov/pubmed). By consulting with several domain experts on the query terms, by researching cancer-related bio articles, and by referring to the cancer gene information in the KEGG database, we selected 37 keywords related to cancer and extracted abstracts from PubMed database only if they contained these keywords. In this way, we collected a total of 2,379,349 abstracts; Figure 2 shows the search query terms used.

In the pre-processing stage, we extracted the PubMed ID, title, and abstract from the PubMed records (which are in XML form) using SAX-based XML parsing module. We also used the Stanford CoreNLP text-mining tool, which includes sentence segmentation, POS tagging, and lemmatization [29].

### 2.2. Named Entity Recognition

Next we identify drug, protein, and side effect entities that mentioned in the text and exist in the databases using dictionary-based Named Entity Extraction [30].

There are several open source NER tools such as MetaMap, Banner, ABNER, and LingPipe. Although machine-learning approaches including Banner, ABNER, and LingPipe are prevalent due to their efficiency, they often suffer from poor performance when applied to real world scenarios. Thus, in our study, because performance accuracy is far more important than efficiency, we constructed an entity dictionary to exactly match entities with sentences from the full text.

#### 2.2.1. Building the Entity Name Dictionary

We constructed the entity name dictionary using consolidated entity names from three publically available databases: Drugbank Version 4.1 (http://www.drugbank.ca/) for drug type entities, Uniprot (http://www.uniprot.org/) for human protein type entities, and SIDER (http://sideeffects.embl.de/) for side effect entities. Finally, we formulated the expanded dictionary to include synonyms of each entry name derived from these databases.

In the case of proteins, multiple synonyms exist for a singular protein, making synonym processing essential. In our research, we used the synonyms that are provided by the UniProt database to construct our synonym dictionary for proteins as well as their synonymous gene names.

#### 2.2.2. Recognition of Entity Name

We used N-gram matching to recognize the entities in sentences we extracted. Sentences are first split and then N-gram tokenized by Apache Lucence (http://lucene.apache.org). N-gram has been performed by 1-, 2-, 3-gram, and these split tokens (token n-gram) are identified within the name entity through three different entity dictionaries. NER for each n-gram is conducted by exact matching.

### 2.3. Relation Extraction

Before merging the three multi-type entity paths, we first focused on the relationship between two entities. If a verb is located between two entities in a sentence and corresponds to the rules that we have made (see below), then we extracted the two entities assuming that they were related, along with the verb. This relation-extraction module is executed to construct three different relations—drug-protein, protein-protein, and protein–side effect—which are then merged to create the entire paths of drug- protein–side effect.

To extract the drug-protein, protein-protein, and protein–side effect relations from sentences that include annotated entities, we prepared and applied these rules to the extraction module with Stanford CoreNLP toolkit.
**Basic rule**: When the sentence structure is presented as entity–verb–entity, we assume that the two entities have a relationship. To identify the main verb in a sentence, we used part-of-speech tagging.**Negation detection:** When a sentence includes negation dependency relation “neg” in the dependency parse tree or when a token matches with the negative word such as “hardly” or “scarcely,” we classified them as negative. We classified all other sentences as non-negative.**Relation keyword**: After implementing part-of-speech tagging, we filtered whether the verb is included in the list of 389 verbs that are commonly used in the biomedical domain, as defined by previous work [31]. We also manually classified the verb lists into two categories: increase (accelerate, enhance, stimulate, activate, etc.) or decrease (inhibit, reduce, abolish, silence, etc.) based on the bio-verb list.**Distance between entities**: The distance, or number of words, between entities linked by a bio-verb (relation keyword) is an important factor in deciding whether the two entities are truly related. We chose a window size of six words, and we extracted two entities and relation keywords only if they are included within the window size.**Direction**: Direction can be determined by the tense (active or passive) using Stanford CoreNLP part-of-speech parsing and dependency parsing results. When “auxpass” dependency relation appears in a given sentence or when the sentence corresponds to the rules such as auxiliary verb + past participle and be verb + past participle, we considered them passive voice; the rest we considered active. If a relation keyword is passive, the direction of verb is reversed: “entity A is activated by entity B” means “entity B activates entity A.”

### 2.4. Path Detection on Drug-Protein–Side Effect

For path analysis, we combined the three different entity relationships generated in the aforementioned stages (Figure 3). Proteins that appear in both pairs of drug-protein and protein–side effect work as a bridge that connects drug and side-effect.

#### 2.4.1. Pair Generation of Drug-Protein

When we merged the heterogeneous path of the three entities, we first selected 50 drug entities related to cancer that appear in the DrugBank database (listed in Table 1). We then detected the drug-protein pairs related to these 50 cancer drugs.

#### 2.4.2. *K*-Protein Depth Path

The path can differ depending on the number of protein entities, as they are bridges connecting drug and side effect. The number of protein entities (*k*) can be presented as the depth of path (*k*-protein). Altering *k* from 1 to 3, we investigated not only the direct paths of drug-protein–side effect, but also the chain reaction of path between proteins. For example, depth 1 is drug-protein1–side effect and depth 3 is drug-protein1-protein2-protein3–side effect.

#### 2.4.3. Path Ranking Algorithm

To rank the integrated paths of drug-protein–side effect, we combined three metrics: i) the lexical co-occurrence–based semantic similarity score COALS [25,26]; ii) the frequency score of the pair’s co-occurrence; iii) the knowledge-based semantic similarity score derived from UMLS [27,28].

First, COALS was used to estimate the similarity between two words with a given text corpus. Basically, it was calculated by the correlation of two co-occurrence vectors both for normalization and for measuring vector similarity. Second, the frequency-weighted score was calculated as the number of occurrences of a pair on k-depth path. From the score of maximum frequency, we calculated the local frequency, based on the specific side effect. Third, we used knowledge-based semantic similarity score determined from a manually developed knowledgebase like UMLS, which gives the general biological relationship between two given entities.

As a linear combination of the three metrics, we ranked the path (α=0.3, β=0.4, γ=0.3).
$$\mathrm{Entity}\;\mathrm{Pair}\;\mathrm{Weight}\;\left(v_{i},\;v_{j}\right)=\alpha \cdot \mathrm{COALS}\left(v_{i},v_{j}\right)+\beta \cdot \frac{\mathrm{LocalFreq}\left(v_{i},v_{j},SE_{k}\right)}{{\mathrm{MaxLocalFreq}}_{SE_{k}}}+\gamma \cdot UMLS\_Semantic\left(v_{i},v_{j}\right)$$
$$\mathrm{Path}\;\mathrm{ranking}\;\mathrm{score}=\overset{n-1}{\underset{i=0}{\sum }}\frac{Entity\;Pair\;Weight\left(v_{i},\;v_{i+1}\right)}{n-1}$$
where $v_{i}$: Certain entity in given path (drug-protein–side effect sequence); ${\mathrm{COALS}}_{v_{i},v_{j}}$: COALS semantic similarity between $v_{i}$ and $v_{j}$, where $v_{i}$ and $v_{j}$ are extracted entities; $\mathrm{LocalFreq}\;\left(v_{i},\;v_{j},SE_{k}\right)$: Frequency between $v_{i}$ and $v_{j}$ among all path of specific side effect; ${\mathrm{MaxLocalFreq}}_{SE_{k}}$: Maximum frequency score of specific side effect among LocalFreq; $UMLS\_Semantic\left(v_{i},v_{j}\right)$: Knowledge-based semantic similarity between $v_{i}$ and $v_{j}$.

## 3. Results

After extracting the relationship between entities from free text using the proposed relation-extraction method, we constructed integrated networks consisting of drug-protein, protein-protein, and protein–side effect pairs. We then identified a series of paths from drug to side effect. Paths were ranked in descending order by the proposed weight function.

### 3.1. Construction of Drug-SE (Side Effect) Path

#### 3.1.1. Extraction of Relation Pairs from Text

The numbers of entity relation pairs extracted from our dataset are in Table 2. There are 1430 kinds of cancer drugs that appeared from the drug-protein pairs extracted from the text, and 2156 types of cancer drug side effects from the protein–side effect pairs. This is 18.47% of all drugs registered in the DrugBank and 33.78% of all the side effects in the dictionary we built based on SIDER.

When limiting our list of target drugs to the previously selected 50 cancer-related drugs and to the relevant side effects, all 50 drugs were listed in drug-protein pairs, while only some of the side effects were extracted from protein–side effect pairs. Related to the selected 50 drugs, all 1988 side effects appeared in SIDER, but only 996 cases were extracted from the text. In other words, we did not extract all the side effects of cancer-related drugs listed in SIDER for the following two reasons: First, 350 side effects were not mentioned in the collected dataset; Second, 543 side effects were not extracted by the dictionary-based side effect extraction, suggesting that our analysis does not cover all cases.

#### 3.1.2. Detection of Drug-Protein–Side Effect Path

To understand the path from drug to side effect, we created paths made of three types of entities by combining drug-protein pairs with protein–side effect pairs. Then, we categorized the paths by *k*, the number of protein existing between drug and side effect (denoted as *k*-protein depth). The suggested prioritizing algorithm ranks each path according to the proposed weight function, and we looked into the top 250 paths with *k* values of 1, 2, and 3. The number of unique drugs, proteins, and side effects are shown in Table 3.

The extracted paths are shown in the form of drug-protein–side effect. However, these include the paths of the drugs that elicited a treatment response, and so include both intended effect and side effect. For example, there were relatively many tumor or cancer cases in side effect section, when a protein and “tumor” were connected by verbs such as “reduce.” As *k* increases, the number of drugs and side effects increase as well. In drug-protein–side effect paths, 17 cases of side effects were shown redundantly, 32 redundant cases of side effects in drug-protein1-protein2–side effect paths, 78 redundant cases of side effects in drug-protein1-protein2-protein3–side effect paths. The number of drugs also increased by 28 in drug-protein–side effect paths, by 34 in drug-protein1-protein2–side effect paths, and by 39 in drug-protein1-protein2-protein3–side effect paths.

#### 3.1.3. Selection of Significant SE Path

The actual side effects and their paths were difficult to track because the prior top 250 paths included both the intended effects of the drugs and their side effects. Therefore, we classified the paths into effects and side effects by considering paths ending with the nodes such as tumor or cancer as effects and excluded it from our extracted paths.

Among the remaining side effect–specific paths, we then selected only significant paths by inspecting extracted verbs. A path was considered significant only if the verbs represent a change to other bio entities. Table 4 shows the final top 20 significant paths ranked according to our ranking function described in Section 2.4.3. We evaluate the top 20 ranked paths. Our work provides a hypothesis as a starting point for new biological research with relatively little time and effort.

### 3.2. Verification of Path

To measure the performance of the proposed method, we examined the top 20 paths to determine whether they were conceptually plausible or not. Because the results covered a variety of biological entities, we also retrieved the sentences deriving entity pair of the path from PubMed in order to prevent the misunderstanding of such a path.

We analyzed the paths and classified them into three types. Type 1 paths involve entities that are related and verbs that are also correctly connected. Type 2 paths involve entities that are related but verbs that are uncertain. And type 3 involves entities of uncertain relation as well as uncertain verbs.

#### 3.2.1. Type Description

We were capable of interpreting type 1 and type 2 paths as shown in Table 5. Although the verbs that connect the entities are not certain in case of type 2, we were able to connect the entities through a literature analysis showing that the entities are highly related. Through our literature analysis, we were able to verify 65% of the 20 paths as plausible. The verified 65% supports our suggested path between drug and side effect. The result of each path is given in Table 5.

There are some cases in which same pair of drug and side effect appear twice, but are connected by different proteins. For example, the top two ranked paths both link the cancer drug anastrozole to the side effect of acute hepatitis.
Type 1: When the entities are related and the verbs that describe the relations are also correctly connected.Type 2: When the entities are related but the verbs that describe the relation are not certain.Type 3: When the relation of entities are not certain and the verbs that describe the relation are also not certain.

#### 3.2.2. Comparison Ranking Functions

For the selected 20 paths, we compared the performance of the proposed method and existing measures: (1) the average betweenness degree using the co-occurrence frequency of two entities, (2) the semantic similarity obtained by utilizing the COALS algorithm, (3) the semantic similarity for the biomedical domain within the framework of UMLS. We judged that the information corresponding to type 1 is a correct path, and that corresponding to type 3 is an incorrect path. We calculated top n-ranked paths and determined how many correct paths exist. We compared the results between the ranking function we proposed and the other functions. We employed precision at k (P@k), a common measurement in the field of information retrieval, for comparison, where P@10 measures the correct answers in top 10 cases. In our analysis, we measured P@5, P@10, P@15, and P@20; the results are shown in Table 6 and Table 7. Table 6 shows the comparison results among the ranking functions in the case that type 1 is the true case and type 2 and 3 are false case whereas in Table 7, type 1 and 2 are the true case and type 3 is the false case. The ranking of extracted 20 paths by P@K shows whether type 1 paths (or type 1 and 2) are ranked high, which implies the ranking algorithm properly predicts the interesting, meaningful paths.

When type 1 is the only true case, the proposed method outperforms the other three methods by 18.75% to 128.92% for P@5-P@20. The second best performance was achieved by COALS, while UMLS performed the worst. The proposed algorithm was particularly outstanding at top 5, and this result is encouraging in cases where researchers want to investigate only a handful of the resulting hypotheses.

When both type 1 and 2 are assumed to be true, the proposed method again outperforms the other three methods, this time by 3.47% to 34.23%. The second best performance was achieved by the co-occurrence method, while UMLS again did the worst.

### 3.3. Example of Literature Analysis

One example for a type 1 path in which the entities are related and the verbs are correctly connected is path 7. Path 7 shows the connection between the drug sorafenib and the side effect dyspepsia. Figure 4 is conceptual model of path 7 that notes the evidence for each entity pair and verb.

#### 3.3.1. Drug-Protein Connection: Sorafenib (Inhibit, Block) p38

Sorafenib is a kinase inhibitor drug that is used to treat primary kidney cancer and advanced primary liver cancer [32,33]. Uncontrolled growth in many cancers is due to a defect in the Ras-Raf-MEK-ERK path, also known as the MAP/ERK path [34]. Sorafenib acts as an inhibitor for several tyrosine protein kinases, such as VEGFR, PDGFR, and Raf family kinases, resulting in the suppression of tumor growth [35]. Researchers have also shown that sorafenib can inhibit the activation of the MAP kinase p38 by a marked decrease in p38 phosphorylation, without affecting total protein levels [36,37]. These findings support the connection between sorafenib and p38, which are linked by the verbs “inhibit” and “block.”

#### 3.3.2. Protein-Protein Connection: p38 (Inhibit) g17

P38 mitogen-activated protein kinases are one of the main subgroups of the MAP kinases that play a vital role in signal transduction, cell differentiation, apoptosis, and senescence [38,39,40,41]. Gastrin-17 (G-17), also known as little gastrin 1, is a form of the protein hormone gastrin that is secreted by the intestine [42]. Gastrin is produced in the G cells of the duodenum and in the pyloric antrum of the stomach, and is released in response to certain stimuli such as hypercalcemia (elevated levels of calcium in the blood) [43,44]. Gastrin stimulates hydrochloric acid/gastric acid secretion by inducing histamine release from ECL cells, functioning as a central regulator for gastric acid secretion [45]. In our study, we combined gastrin-17 with gastrin, which also exists as gastrin-34 and gastrin-14, into a single entity, as all three form of gastrin are produced in the G cells and functions similarly.

#### 3.3.3. Protein-Protein Connection: p38 (Inhibit) Gastrin

We could not find studies directly connecting from p38 to gastrin, although there were studies directly connecting from gastrin to p38. We searched for other factors that can connect from p38 to gastrin and found NF-κB. NF-κB is a protein complex that regulates transcription, cytokine production, and cell survival and is involved in multiple cellular responses [46,47]. Its transcriptional activation was found to be regulated by the p38 MAP kinase activity [48,49,50], which upregulates NF-κB expression through RelA phosphorylation during stretch-induced myogenesis [50]. NF-κB activity was also induced in C2C12 cells by the activation of p38 [51]. IL1B-activated NF-κB downregulated gastrin, and this downregulation occurred both in the presence and absence of IL1B [52,53]. The ectopic expression of the p65 subunit of NF-κB in AGS cells resulted in about nine fold reduction in gastrin levels, suggesting that gastrin is negatively regulated by NF-κB [53]. These findings suggest a connection between p38 and gastrin by way of NF-κB, in which activation of p38 MAP kinase upregulates NF-κB, which represses the transcription of gastrin.

Another factor that can connect p38 to gastrin is calcium. Osteoclasts are a type of bone cell that breaks down bone tissue, a process critical for bone maintenance, repair, and remodeling [54]. Previous research has found that p38 MAP kinase signaling plays a crucial role in PTHrP-induced osteoclastic bone resorption, in which osteoclasts break down bone and result in the transfer of calcium from bone fluid to the blood [55,56]. FR167653, an inhibitor of p38 MAP kinase, was found to inhibit PTHrP-induced osteoclastogenesis in vitro and PTHrP-induced bone resorption in vivo [56]. Studies also show that bone resorption induced by IL-1 and TNF is mediated by p38 MAP kinase and that p38 activity enhances osteoclast maturation and bone resorption in myeloma [57,58]. These findings suggest that p38 MAP kinase activity plays a crucial role in osteoclast maturation and bone resorption, and thereby can regulate calcium levels in the blood. As mentioned, gastrin is released in response to hypercalcemia (an elevated calcium level in the blood), suggesting that p38 can regulate gastrin through calcium. However, more study is needed to understand exactly how p38 regulates calcium.

Through NF-κB, we can support the connection of p38 to gastrin by the verb inhibit; p38 MAP kinase up regulates NF-κB resulting in the inhibition of gastrin. Through calcium, we can support the connection between p38 and gastrin, but cannot support the verb inhibit. However, through our literature research, we have found a high correlation between p38 and gastrin through calcium, suggesting that calcium is likely to be another mediator connecting p38 and gastrin.

#### 3.3.4. Protein-Side Effect Connection: Gastrin (Associate) Dyspepsia

Dyspepsia, also known as indigestion, is a condition in which digestion is impaired. Dyspepsia is highly related to gastrin, as gastrin is a key regulator for the secretion of gastric acid, a digestive fluid formed in the stomach [45]. Dyspepsia can be caused by gastroesophageal reflux disease (GERD), a condition in which stomach acid comes up from the stomach into the esophagus and causes mucosal damage. These findings support the connection between gastrin and dyspepsia by the verb associate.

In short, we suggest dyspepsia as a side effect for sorafenib: sorafenib inhibits p38, thereby inducinggastrin, which results in dyspepsia. Through our method, we suggest a mechanism for how sorafenib can cause dyspepsia. For other analyses of type 2 and 3 paths, refer to the Appendix A.

### 3.4. Direct Link between Drug and Side Effect

The list of the direct links between drug and side effect was extracted from the collected data, and we provided it along with total frequency of drugs and side effects co-occurred in the same abstract in Table 8.

In Table 8, each drug and side effect is accompanied with Concept Unique Identifier (CUI) provided in the UMLS. We extracted total 827 unique links provided in Appendix A. The most frequently co-occurred pairs are Anti-inflammatories and Cachexia and Antioxidants and Cachexia whose the total number of co-occurrences is 17,653.

## 4. Discussion

### Advantage and Significance

Although the side effects of drugs are reported in clinical trials, there are few studies that attempt to explain why. However, there are many studies on the target paths of cancer drugs. Therefore in our research we used text mining to identify the pathways between drugs and side effects. For example, dyspepsia is a known side effect for Sorafenib. However, the entities connecting Sorafenib and dyspepsia were not well known. Through our research, we suggest p38 and gastrin as entities that can connect Sorafenib to its known side effect, dyspepsia. In situations where biological lab research has its limitations, such methods can provide a path between drug and side effect with little time and resources.

In addition, we can suggest a meaningful hypothesis for further research into a drug and its side effects by using our suggested ranking system. Although there could be many paths by which the side effect may occur, our system provides an opportunity to preferentially overview plausible paths.

Lastly, we think that our methodology could provide a more effective prescription of drug. Currently when prescribing drugs, side effects are considered, and the prescription of drugs with harsh side effect is often avoided. By extracting the path between drug and side effect with verbs, our system can help doctors interpret the underlying path. For example, as shown above, p38 and gastrin are related entities that connect Sorafenib to dyspepsia. Through our research, we suggest that Sorafenib may cause dyspepsia by inhibiting p38, thereby inducing gastrin, which may result in dyspepsia. With this information, doctors can assess a patient’s risk of suffering a given side effect, and therefore, more effectively prescribe the drug in question.

## 5. Conclusions

We used text mining to identify possible paths between drugs and side effects by analyzing 2,379,349 research paper abstracts. By extracting the relation between entities from the abstracts’ free text, we suggest detailed mechanisms connecting drugs and side effects.

To make the results suggested by the proposed approach more reliable, we need to apply methods that can increase the accuracy of suggested specific processes such as NER, RE, and path analysis, including a new path ranking algorithm, and also a develop a process for analyzing error cases. Moreover, a method that can consider conditions and situations that affects the relation will be needed in our future work. In addition, since we used a limited number of oncogenes as search queries, it would be desirable to include more comprehensive oncogenes to search queries.

Our proposed framework is not limited to drug–side effect relations, however, and could be applied in various circumstances. For this, it requires first defining a detailed conceptual model; for example, what entity types could be contained in the path. Then, a detailed process such as NER, RE, and path analysis are executed to extract potential entities. Through these pipelines, our suggested method may give meaningful results for biomedical researchers in various situations such as protein-protein interaction (PPI). For PPI, we are able to integrate widely used PPI databases such as Mammalian Protein-Protein Interaction Database (http://mips.helmholtz-muenchen.de/proj/ppi/) and BioGRID (http://thebiogrid.org/) for PPI extraction.

## Figures and Tables

**Figure 1 genes-10-00159-f001:**
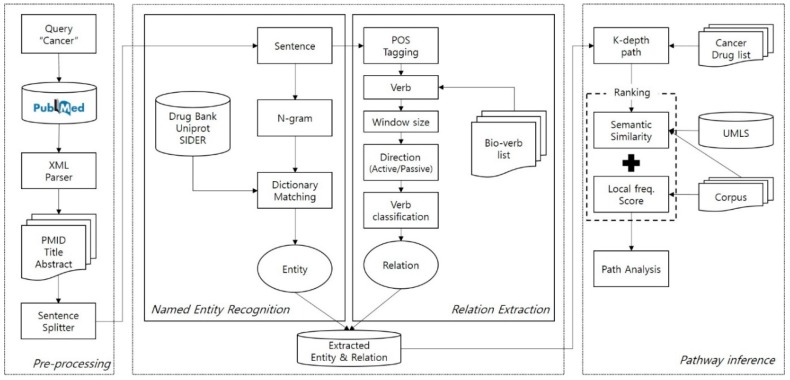
System overview.

**Figure 2 genes-10-00159-f002:**
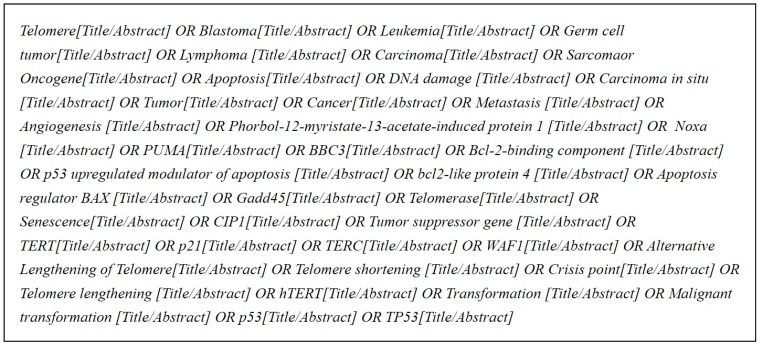
Search query.

**Figure 3 genes-10-00159-f003:**
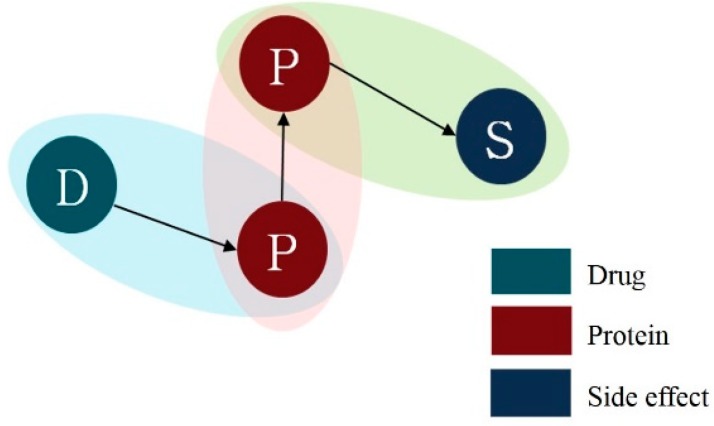
Conceptual model of drug-protein–side effect path. D—drug, P—protein, S—side effect.

**Figure 4 genes-10-00159-f004:**
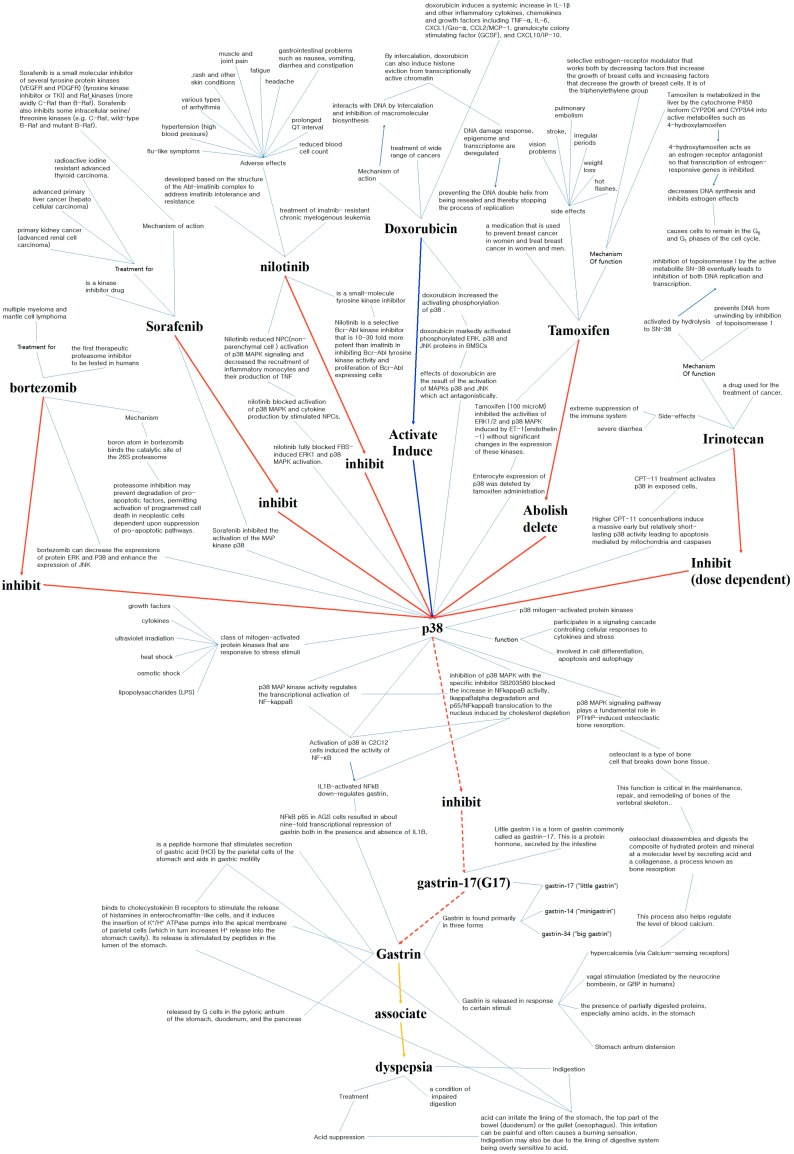
Extracted drug-protein–side effect paths for sorafenib and dyspepsia.

**Table 1 genes-10-00159-t001:** List of selected drugs.

Drug Name	Drug Name	Drug Name	Drug Name	Drug Name
amifostine	Cetrorelix	erlotinib	Letrozole	sorafenib
aminoglutethimide	chlorambucil	exemestane	Leucovorin	tamoxifen
amsacrine	cisplatin	fludarabine	Lomustine	temozolomide
anagrelide	cladribine	flutamide	methotrexate	temsirolimus
anastrozole	clofarabine	gefitinib	Mitotane	thiotepa
bexarotene	dacarbazine	gemcitabine	Nilotinib	topotecan
bicalutamide	daunorubicin	idarubicin	nilutamide	toremifene
busulfan	degarelix	ifosfamide	ondansetron	vincristine
capecitabine	docetaxel	irinotecan	paclitaxel	vinorelbine
carboplatin	doxorubicin	ixabepilone	procarbazine	bortezomib

**Table 2 genes-10-00159-t002:** Number of pairs, source, and target depend on each entity relation.

Coverage	Entity Relation	# of Pairs	# of Unique Source	# of Unique Target
**All drugs**	drug-protein	32,307	1430	2709
	protein-protein	146,912	6125	5904
	protein–side effect	170,112	6222	2156
**50 cancer drugs**	drug-protein	2622	50	1055
	protein–side effect	41,415	5313	996

**Table 3 genes-10-00159-t003:** Number of drug, protein, and SE in top 250 paths.

Path Type	# of Paths	# of Drugs	# of Proteins	# of Side Effects
**drug-protein–side effect (depth-1)**	250	28	131	17
**drug-protein1-protein2–side effect (depth-2)**	250	34	126	32
**drug-protein1-protein2-protein3–side effect (depth-3)**	250	39	222	78

**Table 4 genes-10-00159-t004:** List of top 20 paths.

No	Drug	Verb1	Protein	Verb2	Protein	Verb3	Protein	Verb4	Side Effect
1	anastrozole	observe	ar	decrease	muc5ac	use	polymerase	suggest	acute hepatitis
2	anastrozole	differ	age	associate	muc5ac	use	polymerase	suggest	acute hepatitis
3	irinotecan	inhibit	p38	inhibit	g17	inhibit, neutralize	gastrin	associate	dyspepsia
4	tamoxifen	abolish, delete	p38	inhibit	g17	inhibit, neutralize	gastrin	associate	dyspepsia
5	doxorubicin	induce, activate	p38	inhibit	g17	inhibit, neutralize	gastrin	associate	dyspepsia
6	nilotinib	reduce, increase	p38	inhibit	g17	inhibit, neutralize	gastrin	associate	dyspepsia
7	sorafenib	inhibit, block	p38	inhibit	g17	inhibit, neutralize	gastrin	associate	dyspepsia
8	bortezomib	inhibit, decrease	p38	inhibit	g17	inhibit, neutralize	gastrin	associate	dyspepsia
9	bortezomib	induce	protein kinase	phosphorylate	p150	occur	cd5	present	glomerulonephropathy
10	cetrorelix	reduce, decrease	egf	decrease	p15	increase	smad4	interact	septal defect
11	cetrorelix	inhibit	pcna	conserve	p15	increase	smad4	interact	septal defect
12	bortezomib	induce, stimulate	p53	inactivate	p150	occur	cd5	present	glomerulonephropathy
13	gemcitabine	increase	il-2	stimulate	gls	identify, serve	glutaminase	catalyze	nervous system disorders
14	doxorubicin	decrease, enhance	il-6	interact	clec-2	serve	podoplanin	accelerate	leukoplakia
15	nilotinib	reduce, increase	p38	enhance, inhibit	mao-a	increase	ssao	predict	intracranial hemorrhage
16	cisplatin	induce, increase	tnf-α	increase, decrease	spo	use	lipase	occur	hypophosphatemia
17	methotrexate	inhibit	lp	result	nrl	result, become	rod	lead	nocardiosis
18	chlorambucil	induce, up-regulate	p53	promote	l3mbtl1	enhance, decrease	erythropoietin	exert	ureteral obstruction
19	methotrexate	up-regulate, inhibit	ts	encode	kinesin-2	reduce	rod	lead	nocardiosis
20	methotrexate	separate, observe	cr	increase	cofilin-1	correlate	rod	lead	nocardiosis

**Table 5 genes-10-00159-t005:** Biological analysis result (top 20 path).

Path No.	Drug	Side-Effect	Type1	Type2	Type3
1.	anastrozole	acute hepatitis		O	
2.	anastrozole	acute hepatitis			O
3.	irinotecan	dyspepsia	O		
4.	tamoxifen	dyspepsia	O		
5.	doxorubicin	dyspepsia	O		
6.	nilotinib	dyspepsia	O		
7.	sorafenib	dyspepsia	O		
8.	bortezomib	dyspepsia	O		
9.	bortezomib	glomerulonephropathy			O
10.	cetrorelix	septal defect		O	
11.	cetrorelix	septal defect		O	
12.	bortezomib	glomerulonephropathy			O
13.	gemcitabine	nervous system disorders			O
14.	doxorubicin	leukoplakia		O	
15.	nilotinib	intracranial hemorrhage		O	
16.	cisplatin	hypophosphatemia		O	
17.	methotrexate	nocardiosis			O
18.	chlorambucil	ureteral obstruction		O	
19.	methotrexate	nocardiosis			O
20.	methotrexate	nocardiosis			O

**Table 6 genes-10-00159-t006:** Comparison of precision at *n* (where type 1 = true and both type 2 and 3 = false).

Path Type	Co-Occurrence	COALS	UMLS	Proposed
P@5	0.20	0.40	0.00	0.60
P@10	0.30	0.50	0.20	0.60
P@15	0.40	0.40	0.33	0.40
P@20	0.30	0.30	0.30	0.30

COALS: please define; UMLS: Unified Medical Language System

**Table 7 genes-10-00159-t007:** Comparison of precision at *n* (where both type 1 and 2 = true and type 3 = false).

PATH TYPE	Co-Occurrence	COALS	UMLS	Proposed
P@5	0.80	0.60	0.40	0.80
P@10	0.70	0.90	0.50	0.80
P@15	0.73	0.67	0.67	0.73
P@20	0.65	0.65	0.65	0.65

**Table 8 genes-10-00159-t008:** The top 20 direct links between drug and side effect.

Drug|CUI	Side Effect|CUI	Frequency
ANTI-INFLAMMATORIES|C0003209	CACHEXIA|C0006625	17,653
ANTIOXIDANTS|C0003402	CACHEXIA|C0006625	17,653
NSAIDS|C0003211	LBP|C0024031	7600
PCT|C0032452	LBP|C0024031	7296
SR59230A|C0386264	CACHEXIA|C0006625	3336
PD|C0030230	FATIGUE|C0015672	3234
IRON|C0302583	FATIGUE|C0015672	3234
IBUPROFEN|C0020740	CACHEXIA|C0006625	3197
TG|C0039902	CACHEXIA|C0006625	2919
DIET|C0012155	CACHEXIA|C0006625	2780
ANTIGEN|C0003320	CACHEXIA|C0006625	2502
GEFITINIB|C1122962	FATIGUE|C0015672	2464
TCC|C0077072	FATIGUE|C0015672	2310
SER|C0036720	CACHEXIA|C0006625	2224
ERLOTINIB|C1135135	FATIGUE|C0015672	2156
DTC|C0012194	FATIGUE|C0015672	2002
CNI-1493|C0384938	FATIGUE|C0015672	1848
PROTONS|C0033727	FATIGUE|C0015672	1848
HYDROGEN|C0020275	FATIGUE|C0015672	1848

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
