# Peer review of "Inferring Drug-Protein–Side Effect Relationships from Biomedical Text"

_genes, 2019, doi:10.3390/genes10020159_

Round 1

Reviewer 1 Report

Min Song et al. described a text-mining based technique to connect drugs and their side effects, based on a number of databases (PUBMED, DrugBank, UniProt, SIDER) and NLP tools (Stanford CoreNLP, Apache Lucence, COALS, UMLS). I am very positive for such work that combines text-mining technology with biomedical research. This work could be very useful, if the authors could address the comments below, and make a major revision.

Broad Comments:

The author identifies drug-side effect relationship through linking drug-protein, protein-protein, and protein-side effect relationships from text-mining. However, the direct relationship between drug and side effect is missing (e.g. sorafenib may cause unwanted effects, like headache, nausea, etc.). It is the strongest evidence to link drug and side effect, without mentioning any protein in between.

The best end-product will be a webserver that allows users to search for the associated side effects for a particular drug. If this is too difficult, authors should at least provide a full list of all drug entities (with synonyms, and development status (approved, clinical, investigational)) to their associated side effects, together with the information about the proteins, verbs, path scores, and PMIDs.

It is better to briefly describe the existing studies that extract the relationships between drugs and side effects, in the introduction.

Specific Comments:

Figure 2: the author included a limited number of oncogenes in the 37 keywords related to cancer, for article collection. It’s good to expand the number of oncogenes.

Page 4, line 107: the author should mention if the gene names have been included as the synonyms for each protein? For example, Vascular Endothelial Growth Factor Receptor 2 is a protein name, and its gene names include: VEGFR2, KDR, FLK1.

Page 4, line 134: why choosing a window size of 6 words? Any reference, analysis, or optimization for this point?

Page 5, line 162: It is good to have k-protein depth path, but what is the reason to choose 3? Moreover, the available protein-protein interactions (PPI) extracted from the literatures downloaded based on the 37 keywords could be very limited. For example, drug1 inhibits protein1, and protein2 is associated with side effect1. In the extracted abstracts, protein1 may never be mentioned together with protein2. However, the fact is protein1 forms protein complex with protein2 to activate the following signaling cascade, thus the inhibited protein1 would likely result in side effect1 as well. To improve this part, I may suggest to include the PPI databases that have the description about the nature of the interaction (activate, inhibit, form protein complex, crosstalk, etc.). Therefore, the ‘k-protein depth path’ would be more comprehensive, informative, and predictive.

Page 6, line 174: Why assigning the coefficients (0.3, 0.4, 0.3) to the three score? Is there any optimization or internal validation done here?

Page 6, line 189: Overall, FDA has only approved totally ~2000 drugs, but the author collected 1430 cancer drugs, which seems too many. I assume the author collected approved, clinical trials, and investigational/research drugs in this study. As the drug information was obtained from DrugBank, author should extract the development status (approved, clinical, investigational, terminated) as well, and provide this information in the full list of drug-side effect associations. It can be very helpful to show the side effect profiles of the approved drugs, clinical trial drugs, investigational drugs, and terminated drugs.

Page 6, line 197: As there are only 50% of the side effects extracted from the text, the author should try to adjust the parameters (e.g. word window size) for an improved performance.

Page 7, line 213: I don’t understand why saying the 17 / 32 / 78 side effects are redundant.

Table 4: why are all the top 20 paths having 3-protein depth? The direct drug-side effect relationship should be the most reliable and accurate, as I described previously. I think the more proteins between the drug and the side effect, the less reliability. Additionally, I think the scoring function needs to be optimized.

Table 5: I don’t understand why the type II/III can be ranked as top paths, as the author said: type II has no certain relation, and type III has no certain entities and relation. Also, all the type I paths in this top list are having the same side effect (dyspepsia). Any explanation for this point? Is it biased to the keywords you used to search for literature? Again, the scoring function needs to be optimized.

Author Response

Please refer to the attached response document

Reviewer 2 Report

This paper proposes a text-mining approach for inferring drug-protein-side effect relationships from biomedical literature. The authors used predefined relation-extraction rules to extract three different types of relationships, i.e., drug-protein, protein-protein and protein-side effect. By exploiting the relationships, the authors could construct the paths from drug to protein to side effect. This paper is well written particularly in the method and results part, technically plausible and logically acceptable. However, I have some concerns on other parts which require addressing:

(1) The introduction is insufficient. The authors should enrich the introduction part by including some basic backgrounds, like introducing existing methods for constructing the drug-side effect relationship paths, their advantages and disadvantages, what are the motivations of your method and from which perspectives that your proposed method outperforms the existing ones.

(2) The authors need to more specifically explain each part of Figure 1, i.e., the system overview both in the texts and in the legends. For example, what is the Bio-verb list, UMLS, N-gram, corpus, etc. At least put some captions attached to the figure to explain what they are and refer people to the original papers for details.

(3) I would expect the presentation of comparison results of the proposed method with state-of-the-art methods. Otherwise, it is hard to justify how well the proposed method works.

(4) The authors are expected to provide detailed performance metrics methods to evaluate the proposed method (e.g., something like accuracy, precision, recall, etc). Selecting only some of the correctly identified paths is not sufficient to justify the effectiveness of the method.

(5) The authors should consider establishing a web-server to increase the impact of the proposed method.

Author Response

Please refer to the attached response document.

Round 2

Reviewer 1 Report

I am happy with the revised manuscript and the responses to the comments.

Reviewer 2 Report

The authors have addressed all concerns and I would recommend accepting the paper.